# Functional Characterization of a Novel SMR-Type Efflux Pump RanQ, Mediating Quaternary Ammonium Compound Resistance in *Riemerella anatipestifer*

**DOI:** 10.3390/microorganisms11040907

**Published:** 2023-03-31

**Authors:** Heng Quan, Xiaowei Gong, Wenhui Wang, Fuying Zheng, Yongfeng Yu, Donghui Liu, Qiwei Chen, Yuefeng Chu

**Affiliations:** 1College of Veterinary Medicine, Gansu Agricultural University, Lanzhou 730070, China; quanheng0517@163.com (H.Q.); donghuiliu63@163.com (D.L.); 2State Key Laboratory of Veterinary Etiological Biology, College of Veterinary Medicine, Lanzhou University, Lanzhou Veterinary Research Institute, Chinese Academy of Agricultural Sciences, Lanzhou 730000, China; gongxiaowei@caas.cn (X.G.); zhengfuying@caas.cn (F.Z.); yuyongfengvip@163.com (Y.Y.); chuyuefeng@caas.cn (Y.C.)

**Keywords:** *Riemerella anatipestifer*, SMR efflux pump, quaternary ammonium compound, *GE296_RS02355* gene, RanQ protein

## Abstract

*Riemerella anatipestifer* (*R. anatipestifer*) is a multidrug-resistant bacterium and an important pathogen responsible for major economic losses in the duck industry. Our previous study revealed that the efflux pump is an important resistance mechanism of *R. anatipestifer.* Bioinformatics analysis indicated that the *GE296_RS02355* gene (denoted here as *RanQ*), a putative small multidrug resistance (SMR)-type efflux pump, is highly conserved in *R. anatipestifer* strains and important for the multidrug resistance. In the present study, we characterized the *GE296_RS02355* gene in *R. anatipestifer* strain LZ-01. First, the deletion strain RA-LZ01Δ*GE296_RS02355* and complemented strain RA-LZ01cΔ*GE296_RS02355* were constructed. When compared with that of the wild-type (WT) strain RA-LZ01, the mutant strain Δ*RanQ* showed no significant influence on bacterial growth, virulence, invasion and adhesion, morphology biofilm formation ability, and glucose metabolism. In addition, the Δ*RanQ* mutant strain did not alter the drug resistance phenotype of the WT strain RA-LZ01 and displayed enhanced sensitivity toward structurally related quaternary ammonium compounds, such as benzalkonium chloride and methyl viologen, which show high efflux specificity and selectivity. This study may help elucidate the unprecedented biological functions of the SMR-type efflux pump in *R. anatipestifer*. Thus, if this determinant is horizontally transferred, it could cause the spread of quaternary ammonium compound resistance among bacterial species.

## 1. Introduction

*Riemerella anatipestifer* (*R. anatipestifer*) is a rod or oval-shaped, spore-free, and flagella-free gram-negative bacteria of the genus Riemerella, family Weeksellaceae, order Flavobacterium [1]. The bacteria mainly infect the respiratory tract, skin wounds, or digestive tract of 1–5-weeks-old ducklings and can cause acute or chronic septicemia and serositis in ducks [2]. There are 21 recognized serotypes of *R. anatipestifer*, but there is no cross-protection between them, making vaccination more challenging [3]. Consequently, antibiotics have become an important means for preventing and treating *R. anatipestifer* infections. This would greatly increase the emergence and spread of multiple drug-resistant strains too. In addition, *R. anatipestifer* has a natural resistance to a variety of antibiotics, including aminoglycosides, macrolides, cephalosporins, tetracyclines, lincosamides, amino cyclic alcohols, sulfonamides, and polymyxin [4]. For this reason, it is essential to investigate mechanisms of multidrug resistance (MDR) in *R. anatipestifer* in order to prevent resistance from spreading and resistance from increasing.

The drug efflux pump system is a major mechanism that mediates innate and acquired drug resistance in bacteria. The study of the efflux pump enables us to better understand the structure and function of transporters. Currently, eight multidrug transporter families have been identified [5,6,7,8,9]: the ATP-binding cassette (ABC), the p-aminobenzoyl-glutamate transporter (AbgT), the drug/metabolite transporter (DMT), the multidrug and toxic compound extrusion (MATE), the major facilitator superfamily (MFS), the proteobacterial antimicrobial compound efflux (PACE), the resistance-nodulation-division (RND), and the small multidrug resistance families (SMR). ABS and MFS proteins account for nearly half of the membrane transporters currently known, and most of them export drugs through an electrochemical gradient inwardly directed. Among them, the SMR family proteins are the smallest known MDR transporters found in prokaryotes and they use proton power to transport toxic quaternary cationic compounds (QCCs), also known as quaternary ammonium compounds (QACs) [10].

Despite their small size, SMR proteins confer bacterial host resistance to a broad range of chemically diverse QACs that are used in agriculture, medical, or industrial sectors as cationic surfactants, antiseptics, herbicides, and lipophilic dyes [5]. Benzalkonium chloride (BAC), a common QAC with broad-spectrum biocidal activity and stable properties in short-term and long-term use, is widely used as a surface disinfectant in food processing lines, animal barn environments, health care facilities, and homes [11]. They are also popular ingredients in over-the-counter cosmetics, hand sanitizers, and pharmaceutical products [12]. However, recent studies have shown that the widespread use of disinfectants can promote antibiotic resistance [13]. The SMR efflux systems of gram-negative bacteria have been discovered extensively. These bacteria include *Escherichia coli* (*E. coli*) [14], *Bordetella pertussis* [15], *Klebsiella pneumoniae* [16], *Acinetobacter baumannii* [17], etc. However, the function of the SMR efflux pump in *R. anatipestifer* has not yet been reported.

In this study, we investigated the role of the *RanQ* gene in causing drug resistance in *R. anatipestifer* RA-LZ01 strains and discovered that it confers resistance to QACs. In addition, it has made it possible to better understand how efflux pumps mediate resistance in *R. anatipestifer*.

## 2. Materials and Methods

### 2.1. Bacterial Strains, Plasmids, and Media

A complete list of all bacteria and plasmids used in this study can be noted in Table 1. At 37 °C in 5% CO_2_, RA-GD and LJW-2 strains were grown on tryptic soybean broth (TSB, Oxoid) or tryptic soy agar (TSA, Oxoid) with 5% calf serum. The 2,6-diaminoheptanedioic acid (DPA) of 50 mg/mL is added to Luria-Bertani broth (LB, Oxoid) or LB agar at 37 °C to culture *E. coli X7213*. During this study, antibiotics, detergents, and organic solvents were purchased from Solarbio (Beijing, China). The following antibiotics are used when needed: chloramphenicol (Cm), 30 mg/mL; ampicillin (Amp), 100 mg/mL; erythromycin (Erm), 1 mg/mL; and cefoxitin (Cfx), 1 mg/mL.

### 2.2. Isolation and Culture of Duck Brain Microvascular Endothelial Cells (DBMECs)

The isolation and culture of DBMECs were described in the literature [18] with slight correction. The cerebral cortex of 1-day-old ducklings was excised, the meningeal layer was removed, and the cortex was dissected into small pieces of 0.1 cm, which were cultured in the complete endothelial cell culture medium (Saibaikang (Shanghai) Biotechnology Co., Ltd., Shanghai, China) until the cells grew well around the small pieces. Then, the slices were taken out and the cells were digested with trypsin and cultured in a 6-well plate at a cell density of 10^5^ cells per well. After 24 h, the retroviral vector pLVX-EF1α-SV40-IRES-Puro encoding the open reading frame of SV40 large T antigen was coincubated with the cells. The fused unilamellar cells were obtained by continuous culture in a 6-well plate. The cells were then screened with 2 μg/mL puromycin and those that survived for 3 days were subcultured under puromycin pressure. After 12 generations of passage, a clone group composed of pure DBMECs was obtained, and the immortalized DBMECs were successfully established as an in vitro model of the duck blood–brain barrier.

### 2.3. Cloning of RanQ in Deleted E. coli ATCC35150 for Heterologous Studies

Genome sequence of *E. coli* ATCC35150 (accession number GCA_013168075.1) was retrieved from the GenBank database and used to design guide RNA and an editing template. The *acrB*, *ydhE* and *hsd* genes of *E. coli* ATCC35150 were knocked out using CRISPR/Cas9 technology as described elsewhere with slight modifications [19]. Briefly, we first constructed plasmids pTargetF-*acrB* sgRNA, pTargetF -*YdhE* sgRNA, and pTargetF-*hsd* sgRNA vectors encoding Cas9 endonuclease-guided RNA using primers *acrB* sg, *YdhE* sg, and *hsd* sg, respectively. Then left and right homologous arms of acrAB, ydhE, and hsd genes were amplified from the genome of *E. coli* ATCC35150 using primers *acrAB*-up-F/R, *acrAB*-down-F/R, *ydhE*-up-F/R, *ydhE*-down-F/R, *hsd*-up-F/R, and *hsd*-down-F/R, respectively. The homologous arms were fused by fusion PCR. Then *E. coli* ATCC35150 competent cells containing pCas were prepared. The recombinant plasmid and fusion fragment were electroporated into the competent cells and cultured in LB agar containing kanamycin (50 mg/L) and ampicillin (50 mg/L) at 30 °C overnight. The positive clones were identified by colony PCR. Finally, two plasmids were eliminated to obtain *E. coli* ATCC35150 gene deletion strains Δ*acrB*, ΔydhE, and Δhsd. (Primer sequences are shown in Table 2.)

The *RanQ* is located from nucleotide 511,851 to nucleotide 512,177 in the genome sequence of *R. anatipestifer* RA-LZ01. The putative efflux genes *RanQ* and the entire SMR operon were amplified by a standard PCR protocol using the primer pairs *RanQ*-p-F/*RanQ*-p-R (Table 2), and cloned into the EcoRI and PstI (TaKaRa, Dalian, China) sites of pUC18. The resulting recombinant plasmid p*RanQ* was transformed into deleted *E. coli* ATCC35150 for functional characterization.

### 2.4. Construction of Knockout Strain

In Table 2, we list all primers that were used in this study. Genomic DNA was extracted from LJW-2 strain using the TIANamp bacterial genome extraction kit (TIANGEN, Beijing, China), and the operation method is shown in the instruction manual. Using genomic DNA of LJW-2 strain as template, erythromycin resistance gene Erm was amplified by Erm-F/Erm-R primers. *RanQ*-UpF/*RanQ*-Up-R and *RanQ*-Do-F/*RanQ*-Do-R primers were used to amplify the upstream and downstream homologous arms of *RanQ* gene based on the genomic DNA of the RA-LZ01 strain. Then, the upstream and downstream fragments and Erm resistance gene were fused by fusion PCR to obtain the target DNA fragment of 2001 bp. The fused DNA fragment was linked to the ZERO vector (TransGen, Beijing, China) and sequenced by a professional sequencing company (QINGKE, Xi’an, China) to determine the exact nucleotide sequence. The fusion fragment was digested with restriction endonuclease Sac| and Sph| (TaKaRa, Dalian, China), and then ligated to the pRE112 plasmid similar to restriction endonuclease T4 DNA ligase (TaKaRa, Dalian, China) to produce the recombinant plasmid pRE112-*RanQ*. We selected *E. coli X7213* strains carrying pRE112-*RanQ* on LB agar plates with Cm and DPA (25 μg/mL and 50 μg/mL, respectively). A conjugation procedure was then performed to introduce the recombinant plasmids into the R. anatipestifer RA-LZ01 strain [20]. The positive strains screened on the plate containing 5% calf serum and Erm (1 μg/mL) were identified by primers *OmpA*-F/*OmpA*-R, *RanQ*-F/*RanQ*-R, and *Erm*-F/*Erm*-R. Mutant strains resulting from gene deletions were designated RA-LZ01Δ*RanQ*.

### 2.5. Construction of Complemented Strain

In our previous study, we have referred to the literature and constructed the shuttle expression vector pCPRA using pCP29 [21,22]. Using the parent strain RA-LZ01 as the template, the *RanQ* gene with Pst| and Sph| digested sites was amplified by primer *RanQ*-Co-F/*RanQ*-Co-R and ligated with T4 DNA ligase to the pCPRA vector digested with the same restriction endonuclease to construct the recombinant plasmid pCPRA-*RanQ*. Thw X7213 strain was then transfected with recombinant plasmid pCPRA-*RanQ*. The positive strain X7213-Pcpra-*RanQ* was screened on an LB solid plate containing ampicillin (100 μg/mL) and DPA (50 μg/mL). By conjugation, the pCPRA-*RanQ* plasmid from X7213 was introduced into the RA-LZ01Δ*RanQ* strain [20]. The strains on the TSA plate containing 5% calf serum and CFX (1 μg/mL) were identified by primers RA-*OmpA*-F/RA-*OmpA*-R and *RanQ*-F/*RanQ*-R. The screened positive strain was named RA-LZ01cΔ*RanQ*.

### 2.6. Antimicrobial Susceptibility Testing

To assess the resistance profile of RA-LZ01, deletion mutants, and the corresponding complemented strains, antimicrobial agents were routinely tested with 2-fold serial broth microdilutions to determine their minimal inhibitory concentrations (MICs). The operation procedure is carried out with reference to the literature [22]. We used *E. coli* ATCC 25,922 as a control and antimicrobial agents were present in concentrations ranging from 512 to 0.25 μg/mL.

### 2.7. Quaternary Ammonium Salt Tolerance Assay

To further investigate the tolerance of RA-LZ01, Δ*RanQ*, and cΔ*RanQ* to quaternary ammonium salts, we determined the colony-forming ability of the strains in TSA plates supplemented with different types of quaternary ammonium salts at a final concentration of 3.0 μg/mL, according to the reference [23]. Briefly, the bacteria were cultured in the TSB medium to OD_600_ = 1.0. A suspension adjusted to 10^8^ CFU/mL was serially diluted 10 times to 10^3^ CFU/mL. Then add 6 μ L of each diluted bacterial droplet to the TSA plate with quaternary ammonium salt and dry it. The results were observed after 18 h of inverted culture in a 5% CO_2_ incubator at 37 °C.

### 2.8. Quantitative Real-Time RT-PCR

The strains were inoculated in 10 mL of TSB medium containing half MIC of each quaternary ammonium salt and cultured to OD_600_ = 1.0. The total RNA of the strain was extracted with the bacterial total RNA extraction kit (TIANGEN, Beijing, China), and the RNA concentration was measured by ultraviolet spectrophotometer. Total RNA was reverse transcribed into cDNA using a reverse transcription kit (Vazyme, Nanjing, China) as a template for qRT-PCR. The total volume of the reaction was 20 μL, including 10 μL of 2× Master Mix, 0.4 μL of upstream and downstream primers, 10 ng of cDNA content, and finally water was added to 20 μL. Each sample was in triplicate. Amplification protocol consisted of an initial denaturation step at 95 °C for 30 s and 40 cycles at 95 °C for 10 s and 60 °C for 30 s. The *RecA* gene was used as an internal reference gene [4]. The relative gene expression levels were quantified according to the comparative 2^−ΔΔCT^ method [24].

In addition, in order to evaluate the effects of BAC and MV alone and in combination with efflux pump inhibitors (EPIs) on *RanQ* gene activity, we analyzed the effects of quaternary ammonium salts alone or in combination with inhibitors PAβN (40 μg/mL) and CCCP (5 μg/mL) on *RanQ* gene expression. The relative expression of the *RanQ* gene was assessed by comparing the relative quantity of the respective mRNA in the presence of the BAC and MV, and the quaternary ammonium salt + sub-MIC concentration of the inhibitor, with those of the nonexposed strain. The method is the same as the above operation.

### 2.9. Bacterial Growth Curves

In order to explore whether the deletion of the *RanQ* gene affects the growth of RA-LZ01, we monitored the growth of RA-LZ01, Δ*RanQ*, and cΔ*RanQ*, and recorded the OD_600_ value every 1 h for 20 h [25]. The same method was used to determine the growth of RA-LZ01 when BAC (1 μg/mL) and methyl viologen (MV) (1 μg/mL) were used alone or in combination with the inhibitor PAβN dihydrochloride (PAβN) (40 μg/mL) and Carbonyl cyanide 3-chlorophenylhydrazone (CCCP) (5 μg/mL) to evaluate the role of the *RanQ* gene in the efflux of BAC and MV.

### 2.10. Pathogenicity Test

RA-LZ01, Δ*RanQ*, and cΔ*RanQ* strains were cultured in TSB to logarithmic growth phase, and then their colony forming units (CFU) were measured. Then, the initial concentration of each strain was adjusted to 10^9^ CFU/mL. Each 8-day-old duckling was challenged with an intramuscular injection of 0.1 mL bacterial suspension in the thigh muscle using a standard needle (26 gauge). The ducklings were randomly divided into 4 groups, 10 in each group. The control group was intramuscularly injected with the same amount of sterile saline [25]. After infection, continuous observation was performed for 14 days and the survival curve was drawn.

### 2.11. Bacterial Adherence and Invasion Assay

An adherence assay was conducted on the parent strain RA-LZ01 and the *RanQ* mutant to determine if deletion of the *RanQ* gene would affect adherence of *R. anatipestifer* [26,27]. We grew DBMECs to 95% confluence in 48-well cell culture plates. The cells of each well were infected with 10^8^ CFU of Δ*RanQ* mutant or RA-LZ01 bacteria in an Endothelial Cell Medium (ECM), and incubated for 2 h at 37 °C in an atmosphere containing 5% CO_2_. In order to remove nonadherent bacteria, DBMECs were rinsed with sterile PBS 3 times and then digested with 500 μL 0.25% trypsin. This cell suspension was a 10-fold series diluted with PBS and coated on the TSA panels containing 5 μg/mL polymyxin B to determine the number of viable bacteria. In the invasion experiment, bacterial infection was established and bacteria were counted as described above for the bacterial adherence assay, except that during the 3 h incubation for bacterial infection, the extracellular bacteria were killed by incubating the monolayers with the ECM medium containing ampicillin (100 μg/mL) for another 1 h and washed thrice with PBS. At least three independent assays were conducted on separate days in duplicate.

### 2.12. Biofilm Quantification

An assay involving crystal violet (CV) staining was used to quantify biofilm formation by the RA-LZ01 strain [28]. Briefly, the strains were cultured overnight in TSB. It was then adjusted to 0.1 under OD_655_, and 200 μL was transferred to a 96-well microtiter plate (Corning, NY, USA). The cells were incubated in a 5% CO2 incubator at 37 °C for 6, 12, 24, and 48 h. The cell suspension was discarded and gently washed with PBS 3 times, and then stained with 0.1% CV at room temperature for 30 min. They were then rinsed with distilled water 4 times, dried, and added 100 μL of 95% ethanol to dissolve the crystal violet attached to the wall. The optical density at 595 nm (OD_595_) was determined using a microplate reader (BioTek, Winooski, VT, USA).

The experiment was repeated three times and all samples were measured in triplicate; the mean 1 standard deviation for each sample was calculated from three independent experiments.

### 2.13. Observation under Transmission Electron Microscope

The overnight cultures of RA-LZ01 and Δ*RanQ* were collected after centrifugation at 5000× *g* for 10 min and washed twice with sterile PBS. After the bacterial solution was re-suspended, 10 μL of the bacterial suspension was uniformly coated on the copper net of the formvar film and dried for 10 min. Then, the formvar membrane was stained with 5 μL uranylacetate dihydrate acid for 1 min. Using a transmission electron microscope, the samples were observed and analyzed.

### 2.14. Glucose Metabolism Experiment

The WT strain RA-LZ01, mutant strain Δ*RanQ*, and cΔ*RanQ* were grown until the OD_600_ value was about 1.0, and diluted to 10^8^ CFU/mL with 0.85% sterile saline. The bacterial suspensions (0.08 mL) were added to each trace of the cillin bottles and incubated at 37 °C in a CO_2_ incubator for 24 h. The results were then interpreted based on specifications.

For the triple sugar iron agar test, a well-isolated colony from the nutrient agar plate was taken using an inoculation needle and inoculated in the triple sugar iron agar, first by stabbing to the bottom of the tube through the center of the medium and then by streaking the surface of the slant. The lid was half-plugged and the tube was incubated at 37 °C for 24 h and further observed for any change in the color of agar.

### 2.15. Statistical Analysis

GraphPad Prism version 6.0 Windows software was used to perform the statistical analysis. Students’ *t*-test was used to determine if the difference between the two data sets was significant; a value of *p* < 0.05 was considered significant.

### 2.16. Ethics Statement

All animals were handled by following strict procedures according to the Care and Use of Laboratory Animals published by the Institute of Laboratory Animal Resources (Reference No. LVRIAEC-2020-019). All animal experimental procedures were approved by the Institutional Animal Care and Use Committee of Lanzhou Veterinary Research Institute. 

## 3. Results

### 3.1. Sequence Analysis of RanQ

Bioinformatics analysis of *RanQ* (WP_004920112.1), annotated as a multidrug efflux SMR transporter in *R. anatipestifer* strain RA-LZ01 (NCBI Reference Sequence: NZ_CP045564.1), indicated an open reading frame (ORF) of 327 nucleotides. According to the deduced amino acid sequence, the protein consists of 108 residues and has a molecular mass of 11.59 kDa and a theoretical isoelectric point (pI) of 8.85. The amino acid sequence of *RanQ* alignment, as per the results from protein–protein Basic Local Alignment Search Tool (BLASTP), indicated over 98% identity among different *R. anatipestifer* strains. Based on the predictions of its secondary structure and transmembrane topology, *RanQ* is composed of three helical transmembrane segments, with the N-termini located in the periplasm and the C-termini located in the cytoplasm (Figure 1A). Support for this structure came from an independent analysis that revealed the three-dimensional (3D) structure of the protein (Figure 1B). The predicted product of the *RanQ* gene exhibited low amino acid identity and similarity (24% and 38%, respectively) with other SMR transporters involved in the drug efflux in gram-negative bacteria, such as *E. coli*, Bordetella pertussis, *Klebsiella pneumoniae*, and *Acinetobacter baumannii*. These data indicated that *RanQ* is a new multidrug efflux SMR transporter, which differs from the already-known SMR transporters by the presence of three rather than four transmembrane segments.

### 3.2. Reduced QACs Susceptibility by RanQ

The WT RA-LZ01 and deletion strain Δ*RanQ* were tested for susceptibility to forty-two antimicrobial agents belonging to twelve classes with dissimilar structures. The results showed that the Δ*RanQ* strain significantly increased the resistance to macrolide antibiotics because the strain carried an erythromycin-resistant cassette. Compared with the RA-LZ01 strain, the Δ*RanQ* strain only increased susceptibility to MV (two-fold) and BAC (four-fold), and no change was observed in other QACs (Benzyl dimethyl tetradecyl ammonium chloride (TDBAC), Dodecyl trimethyl ammonium chloride (DTAC), and Benzyl cetyl dimethyl ammonium chloride (HDBAC)). According to these results, the RanQ protein is mainly responsible for QACs resistance (Table 3).

Preliminary experiments showed that RA-LZ01 grew well in TSB medium supplemented with 3 μg/mL MV, BAc, TDBAC, DTAC, and HDBAC. Therefore, we performed spot tests on TSA plates supplemented with 3 μg/mL QACs. The results showed that all strains could form colonies on plates without QACs. RA-LZ01 and cΔ*RanQ* could form colonies in the presence of all QACs, while the deletion strain Δ*RanQ* could only form a small number of colonies in the presence of MV and BAC, and the colony formation efficiency was significantly lower than that of the RA-LZ01 and cΔ*RanQ* strains (Figure 2). These results conveyed that the *RanQ* gene contributed to the resistance of QACs.

### 3.3. Characterization of RanQ in Deleted E. coli ATCC35150

The minimum inhibitory concentrations (MICs) for *RanQ*-harboring cells (deleted *E. coli* ATCC35150/p*RanQ*) were higher (MV (4-fold), BAC (8-fold), DTAC (2-fold), TDBAC (1-fold), and HDBAC (1-fold)) as compared to those for deleted *E. coli* ATCC35150/pUC18 (Table 4). It is significant to note that p*RanQ* alone showed a change in QACs MV and BAC susceptibility as compared to deleted *E. coli* ATCC35150/pUC18.

### 3.4. Up-Regulation of RanQ Gene Transcriptions by QACs

Subinhibitory concentrations (2 μg/mL) of BAc, DTAC, TDBAC, HDBAC, and MV (8 μg/mL) were added to TSB to evaluate whether *RanQ* gene transcription was regulated by antibiotics. Induced by antimicrobial agents, the transcription levels of genes were measured using the qRT-PCR assay. The results indicated that the transcription levels of the *RanQ* gene were up-regulated from 2.5- to 2.8-fold and 2.0- to 2.2-fold (Figure 3A), respectively, after induction with BAC and MV. However, there was no significant difference in the transcription level of the *RanQ* gene in wild geese treated with the other three kinds of QACs. When RA-LZ01 was exposed to BAC and MV, the expression of the *RanQ* gene increased significantly. When PAβN was added, the *RanQ* gene expression did not change significantly. However, the expression of the *RanQ* gene was significantly reduced by the addition of CCCP (Figure 3B).

Error bars indicate standard deviation (n = 3). A representative result is given from the three independent experiments.

### 3.5. RanQ Gene Had No Effect on the Growth of Parent Strains

From the growth curve, When RA-LZ01 was exposed to 1 μg/mL BAC (Figure 4A) and MV (Figure 4B), it did not affect the growth of RA-LZ01. When CCCP was added, the growth of RA-LZ01 was significantly slowed down, whereas no significant change in the growth of RA-LZ01 was observed when PAβN was added (Figure 4). But the growth rate of the deletion strain Δ*RanQ* and the replenishing strain cΔ*RanQ* was not significantly different compared to the RA-LZ01 strain (Figure 5A).

### 3.6. Pathogenicity Test

The results of pathogenicity experiments showed that: On the 14th day, the survival rate of WT RA-LZ01 and supplementary strain cΔ*RanQ* was 20%, and that of deleted strain Δ*RanQ* was 25% (Figure 5B). These data revealed that *RanQ* was not involved in the virulence of the RA-LZ01 strain.

### 3.7. RanQ Gene Is Not Involved in the Biofilm Formation of R. anatipestifer

To determine whether the *RanQ* gene plays a key role in *R. anatipestifer* biofilm formation, the biofilm-forming ability of the parent strain RA-LZ01, deletion strain Δ*RanQ*, and complement strain cΔ*RanQ* were investigated at 6 h, 12 h, 24 h, and 48 h. Although there was a difference in the biofilm formed at 6 h by the deleted strain and the parent strain, no significant differences were observed at 12 h, 24 h, and 48 h among the three strains (Figure 5C). This confirmed that the *RanQ* gene was not involved in the biofilm formation of *R. anatipestifer*.

### 3.8. Effect of Deletion of RanQ gene on the RA-LZ01 Morphology of WT

Using a transmission electron microscope, we observed the morphology of the RA-LZ01 and Δ*RanQ* strains. After negative staining with uranyl acetate, the WT strain RA-LZ01 and mutant strain Δ*RanQ* were observed microscopically to have capsules that completely encompass the cells and exhibit rugose formation (Figure 5D). These data implied that deletion of the *RanQ* gene does not affect the RA-LZ01 morphology of WT.

### 3.9. Adhesion and Invasion Assay

The adhesion and invasion experiments showed that there was no significant difference in the adhesion and invasion ability of the deleted strain Δ*RanQ* to DBMECs as compared with that of WT RA-LZ01 and cΔ*RanQ* (Figure 6A,B). Thus, it can be inferred that *RanQ* was not involved in the interaction between *R. anatipestifer* and the host cells.

### 3.10. Effect of Deletion of RanQ Gene on the RA-LZ01 Glucose Metabolism

It was found that RA-LZ01 could utilize glucose, fructose, and maltose, but not lactose and fructose. The deletion of the *RanQ* gene did not affect the glucose metabolism of WT RA-LZ01 (Figure 7A). The triple sugar iron agar test showed a color change from purplish red to yellow, indicating that all three strains could produce acid and make the culture medium yellow (Figure 7B).

## 4. Discussion

In gram-negative bacteria, SMR efflux pumps play a significant role in MDR. Drug transport in the SMR family requires proton exchange to help drive drug efflux through a series of conformational changes. So far, the reporter *EmrE* in *E. coli* has been the most studied in gram-negative bacteria [29], followed by *Acinetobacter baumannii* [17], *Klebsiella pneumoniae* [16], *Bordetella pertussis* [15], etc. In this study, the genome sequence analysis suggested that the *RanQ* gene in *R. anatipestifer* RA-LZ01 encodes a putative SMR efflux pump, and further investigations on the biological function of *RanQ* in *R. anatipestifer* RA-LZ01 were carried out.

By adding efflux substrate, the expression of efflux pump genes was induced, such as the raeB gene in R. anatipestifer CH-1 [30]. The parent strain RA-LZ01 was induced by TDBAC, DTAC, HDBAC, MV, and BAC, and then fluorescence quantitative real-time RT-PCR was performed. We found that the transcriptional level of the *RanQ* gene was up-regulated by 1.1–2.8 times, indicating that these QACs were the efflux substrates of the RA-LZ01 strain *RanQ* gene.

It has been reported that multidrug efflux pumps play a role in the diverse phenotypes of bacteria, including growth and virulence, such as *MmpL11* in *Mycobacterium tuberculosis* [31], *AcrB* in *Salmonella typhimurium* [32], and *AcrB* in *Klebsiella pneumoniae* [33]. However, the deletion of the *RanQ* gene did not affect the growth and virulence of RA-LZ01, indicating that the *RanQ* gene was not involved in the growth and virulence of *R. anatipestifer*. Further the adhesion and invasion abilities of the mutant strain to DBMECs were analyzed and the results manifested that the adhesion and invasion capacities of the Δ*RanQ* were not significantly varied compared to those of the WT strain RA-LZ01. At the same time, biochemical experiments showed that the *RanQ* gene did not play a role in the glucose metabolism of *R. anatipestifer*.

As another phenotype indicating bacterial pathogenicity, biofilms are communities of microorganisms surrounded by extracellular polymeric matrixes. Numerous pathogens, such as *E. coli*, *Streptococcus suis*, and *Haemophilus influenzae*, form biofilms [34]. A biofilm provides bacterial protection from biotic and abiotic stresses in most hostile environments [35]. As part of our previous analysis of our *R. anatipestifer* collection, we found that RA-LZ01 does not produce a strong biofilm. The disruption of *RanQ* does not affect the formation of biofilms, as we found in the biofilm quantification experiment. The pathogenicity of ΔRanQ strain was not significantly different from that of the wild strain RA-LZ01, which was consistent with the observation that the mutant strain could not enhance the biofilm formation of the strain.

The importance of antimicrobial compounds has been highlighted recently due to the COVID-19 pandemic outbreak [36]. QACs are antiseptics that have shown a wide spectrum of antibacterial activity. Many studies have been carried out involving the applications of QACs as antifouling agents for the inhibition of biofilm growth on medical implants and as antibacterial agents on surfaces and aquatic environments [37]. Since the outbreak of COVID-19, QACs are the most widely used disinfectants. The expression of *RanQ* in the heterologous host *E. coli* ATCC35150 caused decreased susceptibility to several agents, such as MV and BAC. It is suggested that the QACs resistance mediated by the efflux pump should be taken seriously.

Efflux pumps are transmembrane transporters, expressed in all types of prokaryotes and eukaryotes with wide substrate specificity that have common amphipathic character and ionizable groups [38]. The SMR efflux pump mediates resistance to antibiotics and compounds such as acriflavine, BAC, Betaine, chloramphenicol, cetyl trimethylamine, crystal violet, ethidium bromide, MV, tetraphenyl phosphine chloride, and so on. In this study, the MIC values of the strains to different antibiotics were determined, and it was found that the deletion strain Δ*RanQ* increased susceptibility to QACs MV and BAC, thereby indicating that the *RanQ* gene mediated resistance to QACs in the RA-LZ01 strain. Our observations corroborated well with the previously defined functions for SMR-type efflux pumps, such as in *E. coli emrE* [14] (betaine and choline), *Acinetobacter baumannii AbeS* [17] (deoxycholate, sodium dodecyl sulfate, acriflavine, BAC), *Bacillus subtilis EbrAB* [39] (ethidium bromide), and *Staphylococcus aureus QacG* [40] (acriflavine, BAC, Cetyl trimethylamine, and ethidium bromide). Reports on the role of SMR-type efflux pumps in antimicrobial resistance have been demonstrated in a few clinically significant pathogens; however, its function has never been investigated in *R. anatipestifer*, thus far. All these data suggest that SMR-type efflux pumps have multiple different physiological functions in different species of bacteria. Overall, we report here for the first time the single biological characterization of an SMR-type *RanQ* efflux system for specific efflux of QACs in *R. anatipestifer*.

## Figures and Tables

**Figure 1 microorganisms-11-00907-f001:**
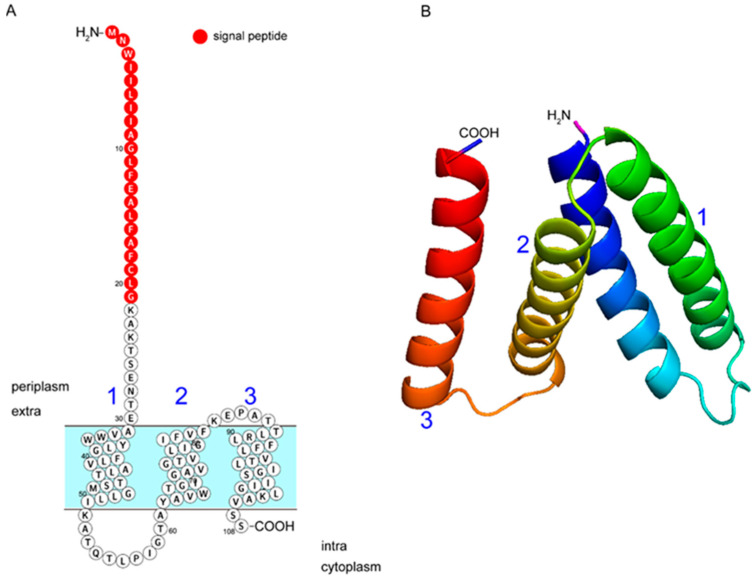
*RanQ* secondary structure and transmembrane topology. (**A**) Prediction of the structure of *RanQ* using Protter. The putative protein is shown parallel to the cytoplasmic membrane. (**B**) 3D representation of *RanQ* viewed along the plane of the membrane from the periplasmic side and visualized using the PyMOL 2.5 software. The 3 transmembrane α-helices are numbered (1 to 3); signal peptides are marked with red circles; and the N-termini is in the periplasm while the C-termini is in the cytoplasm.

**Figure 2 microorganisms-11-00907-f002:**
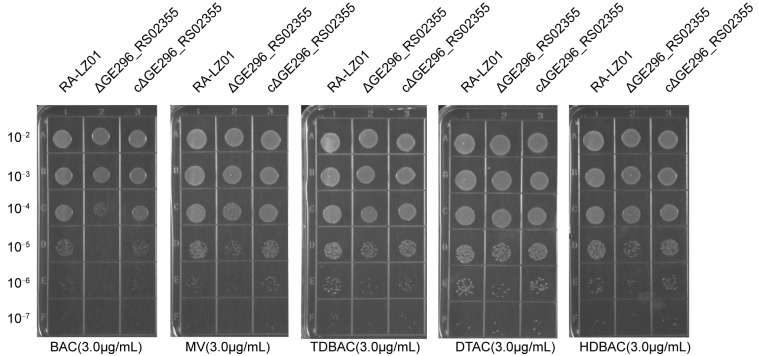
Comparative analyses of QACs resistance in parent strain RA-LZ01 and mutant strain. Growth of parent strain RA-LZ01, deletion strain RA-LZ01Δ*ranQ*, and complemented strain RA-LZ01cΔ*RanQ* were determined by spot assay on TSA plates containing the same concentration of QACs. Remarkably, the deletion strains were grown on TSA plates including 5% calf serum, and a significant difference was observed in the presence of BAC and MV. A representative result is given from the three independent experiments.

**Figure 3 microorganisms-11-00907-f003:**
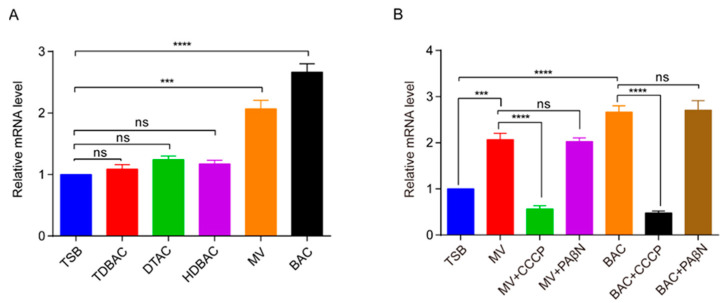
Relative fold changes of *RanQ* mRNA expression levels in *R. anatipestifer* RA-LZ01 after treatment with QACs and EPIs. The subinhibitory concentrations of QACs were added to the TSB broth (TDBAC, 2 μg/mL; DTAC, 2 μg/mL; HDBAC, 2 μg/mL; MV, 8 μg/mL; BAC, 2 μg/mL). TSB—tryptic soybean broth. (**A**) Quantification of the relative mRNA expression levels of the *RanQ* gene in the RA-LZ01 strain. (**B**) Relative fold changes of transcriptional level of the virulence experiment of *R. anatipestifer* WT RA-LZ01, deletion mutant Δ*RanQ*, and complemented strain cΔ*RanQ* infecting ducklings. Gene induced by specific substrates in the parental strain. Transcriptional levels of the virulence experiment of *R. anatipestifer* WT RA-LZ01, Δ*RanQ*, and cΔ*RanQ* infecting ducklings. (ns *p* > 0.05, *** *p* < 0.001, **** *p* < 0.0001).

**Figure 4 microorganisms-11-00907-f004:**
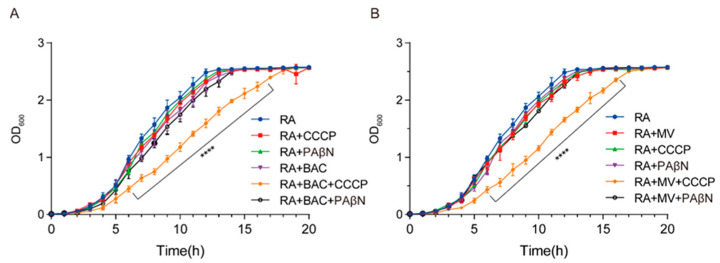
Quantitative susceptibility testing of BAC and MV for the *R. anatipestifer* strains, in the presence or absence of each EPIs. (**A**) Growth curve of RA-LZ01 strain exposed to BAC in the presence of efflux pump inhibitors. (**B**) Growth curve of RA-LZ01 strain exposed to MV in the presence of efflux pump inhibitors. Growth curves of RA-LZ01 in TSB, TSB + CCCP, TSB + BAC + CCCP, TSB + BAC + PAβN, TSB + PAβN, TSB + MV + CCCP, and TSB + MV + PAβN. Each point represents the mean ± standard for triplicate samples. (**** *p* < 0.0001).

**Figure 5 microorganisms-11-00907-f005:**
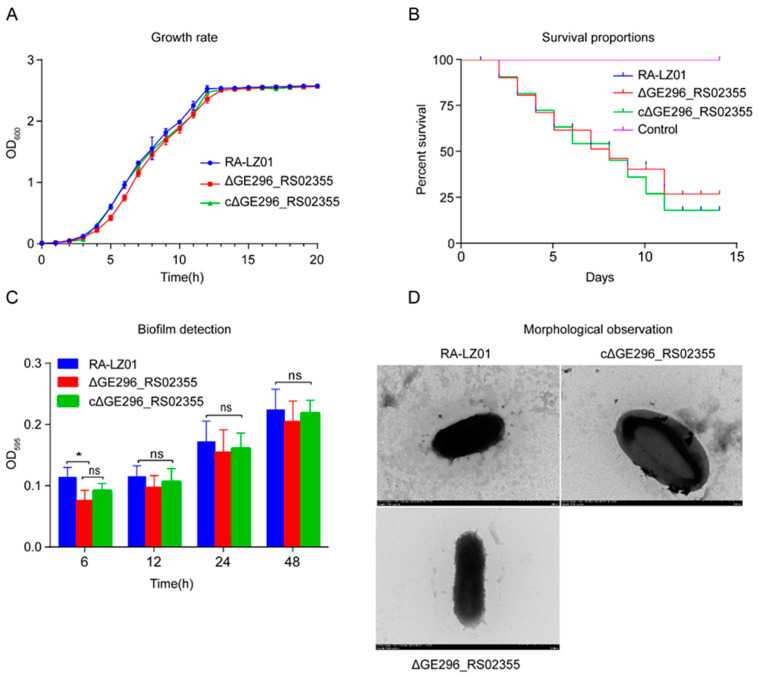
Characterization of parent strain RA-LZ01 and mutant strain. (**A**) Growth curves of RA-LZ0, Δ*RanQ*, and cΔ*RanQ* in TSB. (**B**) The virulence experiment of *R. anatipestifer* WT RA-LZ01, Δ*RanQ*, and cΔ*RanQ* infecting ducklings. (**C**) The biofilm formation ability of *R. anatipestifer* WT RA-LZ01, Δ*RanQ*, and cΔ*RanQ* were analyzed. Each point represents the mean ± standard for triplicate samples. (* *p* < 0.05, ns *p* > 0.05). (**D**) Morphological observation of cells.

**Figure 6 microorganisms-11-00907-f006:**
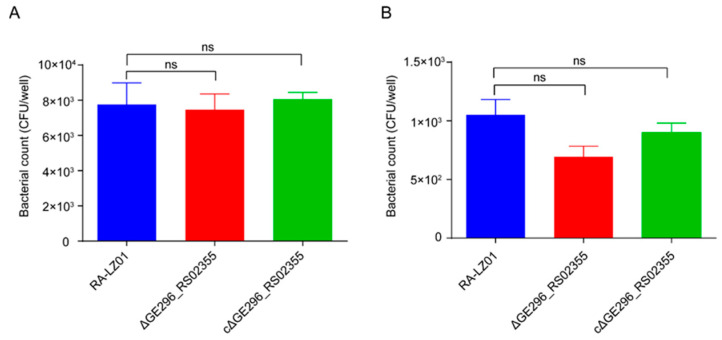
Bacterial adherence and invasion assays. (**A**) Adherence assay. (**B**) Invasion assay. Strains RA-LZ01, RA-LZ01Δ*ranQ*, and RA-LZ01cΔ*RanQ* were tested on Strains RA-LZ01, RA-LZ01Δ*RanQ*, and RA-LZ01cΔ*RanQ* using duck brain microvascular endothelial cells. The data indicate the number of bacteria adherence to or invasion of duck brain microvascular endothelial cells per well in 48-well plates. The standard deviation is calculated from three independent experiments and expressed by error lines. (ns, *p* > 0.05).

**Figure 7 microorganisms-11-00907-f007:**
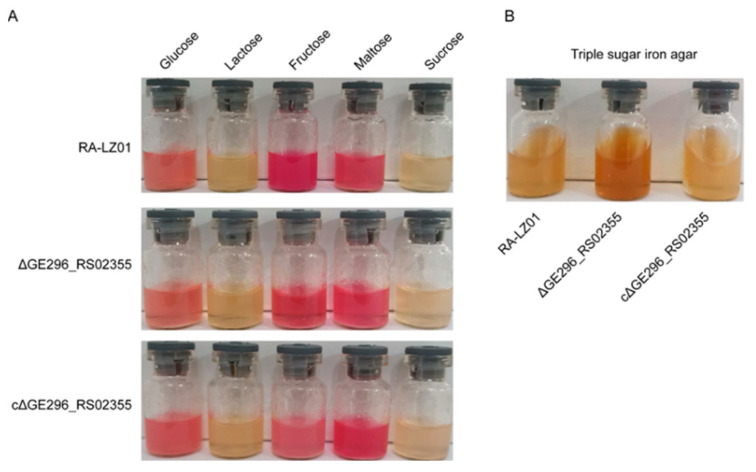
Biochemical experiments of parental strain RA-LZ01, deletion strain RA-LZ01Δ*RanQ*, and supplement strain RA-LZ01cΔ*RanQ*. (**A**) Glycometabolism. (**B**) Triple Sugar Iron assay.

**Table 1 microorganisms-11-00907-t001:** Bacterial strains and plasmids used in this study.

Strains and Plasmids	Description	Source
Strains		
*R. anatipestifer* RA-LZ01(GenBank Accession number: CP045564.1)	Serotype 1, Erm^S^, Cm^S^, Kan^R^	This study
*R. anatipestifer* LJW-2	Serotype 2, Cm^S^, Erm^R^, Kan^R^	This study
RA-LZ01Δ*RanQ*	Serotype1, Cm^S^, Erm^R^, Kan^R^	This study
RA-LZ01cΔ*RanQ*	Serotype1, Cm^S^, Erm^R^, Kan^R^ Cfx^R^	This study
*E. coli* X7213	Thi21 thr21 leuB6 fhuA21 lacY1glnV44ΔasdA4 rexA1 RP4 22Tc::Mu [λpir] Km^R^	BioVector NTCC Inc. (Beijing, China)
*E.coli* X7213-pRE112::Erm-600 bp	*E. coli* X7213 carrying pRE112::Erm-600 bp. Cm^R^, Erm^R^	This study
*E. coli* X7213-pCPRA::*RanQ*	*E. coli* X7213 carrying pCPRA::*RanQ*. Amp^R^, Cfx^R^	This study
*E. coli* ATCC35150	Pathogen: clinical or host-associated sample from *Escherichia coli* O157:H7	BioVector NTCC Inc.
Plasmids		
pRE112	sacB mobRP4 R6K ori, pRE112-T-vector, Cm^R^	BioVector NTCC Inc.
pRE112::Erm-600 bp	The pRE112 plasmid carrying the target fragment contained the upstream and downstream homologous arms of the erythromycin gene and *RanQ* gene. Erm^R^, Cm^R^	This study
pCP29	Amp^R^, Cfx^R^	BioVector NTCC Inc.
pCPRA::*RanQ*	Recombinant plasmid pCPRA carrying *RanQ* gene. Amp^R^, Cfx^R^	This study
pCas	Kan^R^	BioVector NTCC Inc.
pTargetF	Spe^R^	BioVector NTCC Inc.

Note: R, resistance; S, sensitive; Amp, ampicillin; Erm, Ergomycin; Cm, chloramphenicol; Kan, kanamycin; Cfx, cefoxitin; Spe, spectinomycin.

**Table 2 microorganisms-11-00907-t002:** This study used qRT-PCR and PCR primer.

Primers	Sequences (5′-3′)	Amplicons (bp)	Source
*recA*-F	ATTGATGGTGATATGGGAGAT	157	This study
*recA*-R	CAGGGCTACCAAACATTACTC		
*RanQ*-Q-F	CCGGAATAGGAGCAGTAGGC	74	This study
*RanQ*-Q-R	GAGCCTTAGTGTGGTAGCGG		
*RanQ* -Up-F	CGAGCTCATCGCATGTACTGGTACTGGTGGAA	627	This study
*RanQ* -Up-R	GGCAATTTCTTTTTTGTCATCTTAAATGGTATAAAAAAGG		
*RanQ* -Do-F	AAAAATTTCATCCTTCGTAGCAATCATCAATTTAATTCTC	627	This study
*RanQ* -Do-R	GGCATGCAAAGTCAGTTTCACAAAAGAGTAAA		
*Erm*-F	CTACGAAGGATGAAATTTTTC	801	This study
*Erm*-R	ATGACAAAAAAGAAATTGCCCG		
*OmpA*-F	ATGTTGATGACTGGACTTGGT	1119	This study
*OmpA*-R	CTTCACTACTGGAAGGTCAGA		
*RanQ* -F	ATGAATTGGATTATTTTAATCATTG	327	This study
*RanQ* -R	TTAACTTGATACAGCTTTTAGCCCG		
*RanQ*-p-F	GGAATTCATGAATTGGATTATTTTAATC	341	This study
*RanQ*-p-R	GCTGCAGTTAACTTGATACAGCTTTTAGCCCG		
*acrB* sg	AAGCGACGCTTGATGCGGTG	20	This study
*YdhE* sg	GATCTGGTCGTTGAACCTAT	20	This study
*hsd* sg	GTTCGGAAGTAATATCACAA	20	This study
*acrB*-up-F	GGGGCAAAGAGCCAGTTTTCCATC	686	This study
*acrB*-up-R	TAGTGATTACACGTTGTAATGTAAACCTCGAGTGTCCGAT		
*acrB*-down-F	GACACTCGAGGTTTACATTACAACGTGTAATCACTAAGGC	778	This study
*acrB*-down-R	TCGTCGATCTGCTCAATGAGCTTA		
*ydhE*-up-F	GGTGACAGTGTCACTTTCAGTAT	334	This study
*ydhE*-up-R	CAAATAAAAGGTGTTCACTAAAGACAAGGCGCAACCTTCA		
*ydhE*-down-F	GGTTGCGCCTTGTCTTTAGTGAACACCTTTTATTTGTAGT	435	
*ydhE*-down-R	CCAAGATTGGTAATGCGCAACGT		This study
*hsd*-up-F	GAAAGCGGAGTCGATCGTTACTT	330	
*hsd*-up-R	TACCGGTTCGTTAGTGTAATAAACCTCCTGTGAACTTCAG		
*hsd*-down-F	AGTTCACAGGAGGTTTATTACACTAACGAACCGGTAAACAG	437	This study
*hsd*-down-R	GCCCTCTCCTGGTCCTGTAAGAT		

**Table 3 microorganisms-11-00907-t003:** An evaluation of MIC values for various antimicrobials in the presence of RA-LZ01, RA-LZ01Δ*RanQ*, and RA-LZ01cΔ*RanQ* strains.

Antimicrobial Category	Antibiotic	Strain		
		RA-LZ01	Δ*RanQ*	*c*Δ*RanQ*
Macrolides	Roxithromycin	0.25	512	512
	Erythromycin	0.25	512	512
Aminoglycosides	Neomycin sulfate	256	256	256
	Amikacin	256	256	256
	Tobramycin	512	512	512
	Gentamicin	128	128	128
	Kanamycin	512	512	512
	Streptomycin	512	512	512
Tetracyclines	Oxytetracycline	8	8	8
	Doxycycline	1	1	1
	Tetracycline	4	4	4
Quinolones	Enrofloxacin	<0.25	<0.25	<0.25
	Ciprofloxacin	<0.25	<0.25	<0.25
	Norfloxacin	<0.25	<0.25	<0.25
Chloramphenicol	Chloramphenicol	0.5	0.5	0.5
	Florfenicol	0.25	0.25	0.25
	thiamphenicol	1	1	1
Sulfonamide	Sulfamethoxine	>512	>512	>512
	Sulfa-p-oxypyrimidine	>512	>512	>512
	Sulfamethoxazole	>512	>512	>512
Polymyxins	Polymyxin B	>512	>512	>512
	Colistin E	>512	>512	>512
Detergent	SDS	8	8	8
	Triton X-100	4	4	4
Quaternary ammonium cation	MV	64	32	64
	BAC	16	4	16
	TDBAC	4	4	4
	DTAC	8	4	8
	HDBAC	4	4	4

**Table 4 microorganisms-11-00907-t004:** MICs of various antimicrobial agents for deleted *E. coli* (ATCC35150/pUC18 and ATCC35150/pUC18-*RanQ*) strains used in this study.

Compound	MIC(μg/mL) for Strains				
	*E. coli*				
	ATCC35150	ATCC35150 ^a^	ATCC35150/pUC18	ATCC35150/pUC18-*RanQ*	Fold Change ^b^
Amikacin	2	0.125	0.125	0.125	1
Chloramphenicol	8	0.5	0.5	0.5	1
Tetracycline	4	0.125	0.125	0.125	1
Chlorhexidine	0.5	0.0625	0.0625	0.0625	1
Polymyxin B	0.125	0.125	0.125	0.125	1
Methyl Viologen	8	2	2	8	4
Benzalkonium chloride	16	1	1	8	8
TDBAC	16	1	1	1	1
DTAC	8	1	1	2	2
HDBAC	8	1	1	1	1

(a) *E. coli* ATCC35150 with deletion of *acrB*, *ydhE*, and *hsd* genes. (b) Fold change is the ratio of MICs for Puc18-*RanQ* to pUC18.

## Data Availability

All the data were included in the manuscript.

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
