# Peer review of "Functional Characterization of a Novel SMR-Type Efflux Pump RanQ, Mediating Quaternary Ammonium Compound Resistance in Riemerella anatipestifer"

_microorganisms, 2023, doi:10.3390/microorganisms11040907_

Round 1
Reviewer 1 Report
The manuscript describes and characterizes the gene encoding efflux pump specific for antibiotics in Riemerella anatipestifer. The manuscript is well structured, and the experiments are performed very systematically, especially the method for the construction of knock-out mutants is interesting since the CRISPR-Cas9 system has not been often used for this purpose in bacteria, yet. I support the manuscript for publication. I have two minor suggestions:
- the bacterium does not belong to the genus Flavobacterium, the genus Riemerella. Maybe the authors can mention family.
- L76-78: This information is better included in an additional column of Table 1.
Reviewer 2 Report
In this work by Quan et al., 2023, the authors investigated the biological function of the RanQ efflux pump in Riemerella anatipestifer challenged with Quaternary Ammonium and other antimicrobial agents. The manuscript is well-written, and the methods and results are presented logically and convincingly. While the work is relevant, some methodological points need to be better detailed before the publication of this manuscript. Please see below:
1) lines 57-62: this whole period is under-referenced.
2) line 71: please remove the excessive space.
3) line 92: Was the brain kept in a physiological solution during tissue fractionation, or was it dry-sectioned?
4) line 93: please, to be clear, no other substances were added to the medium, like the fetal bovine serum, antibiotics, etc.
5) line 95: Did the plates need to be treated to allow cell adhesion?
6) line 95: What is the trypsin concentration, and how long was it kept in culture?
7) line 90: What was the temperature and atmosphere of incubation of the cells?
8) lines 97-101: how often the selective pressure medium containing puromycin was changed? Was the puromycin concentration used during the 12 passages the same as in the first challenge (2 μg/mL)?
9) line 90: Please explain in detail the process of obtaining the primary cultures to allow the reproducibility of the study. If necessary, include the reference that subsidizes the procedure.
10) line 181: what is the name of the kit used for the conversion into cDNA?
11) line 207 and 211: what is the concentration of the saline solution used? What centrifugation was used to prepare the inoculum in saline solution?
12) line 250: Please include a space.
13) line 329: please put b instead of B.
14) line 440: please put the scientific names in italics.
15) lines 449-452: this period is under-referenced.
